# Healthy cloned offspring derived from freeze-dried somatic cells

Sayaka Wakayama [1,2✉], Daiyu Ito [1], Erika Hayashi [1], Takashi Ishiuchi[1] & Teruhiko Wakayama [1,2✉]

Maintaining biodiversity is an essential task, but storing germ cells as genetic resources using liquid nitrogen is difficult, expensive, and easily disrupted during disasters. Our aim is to generate cloned mice from freeze-dried somatic cell nuclei, preserved at −30 °C for up to 9 months after freeze drying treatment. All somatic cells died after freeze drying, and nucleic DNA damage significantly increased. However, after nuclear transfer, we produced cloned blastocysts from freeze-dried somatic cells, and established nuclear transfer embryonic stem cell lines. Using these cells as nuclear donors for re-cloning, we obtained healthy cloned female and male mice with a success rate of 0.2–5.4%. Here, we show that freeze-dried somatic cells can produce healthy, fertile clones, suggesting that this technique may be important for the establishment of alternative, cheaper, and safer liquid nitrogen-free bio-banking solutions.

[1] Faculty of Life and Environmental Science, University of Yamanashi, Kofu 400-8510, Japan. [2] Advanced Biotechnology Center, University of Yamanashi, Kofu 400-8510, Japan. ✉email: sayakaw@yamanashi.ac.jp; twakayama@yamanashi.ac.jp

The preservation of genetic resources is an important tool in promoting species survival. Although not all genetic traits are required for survival, they must be preserved so that species can survive if unknown diseases are spread or environmental changes, such as global warming, occur[1]. Spermatozoa and embryos of mammals are cryopreserved in liquid nitrogen (LN2)[2,3]. However, the use of LN2 has several drawbacks, such as high maintenance cost, and during a disaster, LN2 supply may be stopped or supplies may be destroyed. To solve this problem, we developed a freeze-drying technique for mouse spermatozoa[4]. Although all spermatozoa died after the freeze-drying process, their DNA remained intact, and healthy offspring were obtained when spermatozoa were injected into oocytes. This technique has also been applied to other species, such as rats, hamsters, rabbits, and horses[5–8]. The nuclei of freeze-dried (FD) mouse spermatozoa have a strong tolerance against environmental changes[9]; healthy offspring were obtained from FD sperm stored for more than one year in a desk drawer without controlling room temperature[10] and in the International Space Station for more than 5 years[11,12]. Thus, freeze drying could be the best way to preserve genetic resources for a long period in a safe, low-cost, and location-independent manner[13,14]. However, to date, the only cells that have produced offspring after freeze drying are mature spermatozoa. Collecting spermatozoa from infertile males and oocytes/embryos from fertile females is difficult. Conversely, after the success of the first animal (frog) clone[15,16] and the first mammalian (sheep) clone[17], it is now possible to generate cloned offspring from live somatic cells[18]; this indicates that somatic cells can also be used as a genetic resource. It is noteworthy that somatic cells can be collected from almost anywhere in the body or even from body waste[19] or cadavers[20,21].

In this study, we aimed to generate cloned mice from FD somatic cell nuclei by adapting the nuclear transfer procedure. Our data reveal that although some DNA abnormalities are observed in the process, FD somatic cell nuclei can be used to generate blastocysts by nuclear transfer, and embryonic stem cell lines derived from these blastocysts yield donor nuclei that are capable of producing healthy, fertile cloned mice. The results obtained in this study may provide a viable method for preserving the genetic resources of any animal in a safe and low-cost manner, even if power and LN2 supplies are interrupted during a disaster.

## Results

**Morphology of somatic cells after FD treatment.** Somatic (cumulus) cells were freeze-dried with trehalose as a cryoprotectant or epigallocatechin as an antioxidant (Figs. 1 and 2a). The rehydrated FD somatic cells treated with epigallocatechin were more three-dimensional and rounder (Supplementary Fig. 1), compared to the FD somatic cells treated with trehalose. However, all cells were positive for propidium iodide (PI) staining irrespective of the treatment type (Fig. 2b), indicating that the membranes of all FD somatic cells were broken. We cultured the somatic cells in an incubator for 1 week, and none of them were attached to the bottom of the dish (Supplementary Fig. 2).

**DNA damage of donor cell nuclei.** The comet DNA breakage assay (Fig. 1 "1. Comet") was used to detect DNA damage in FD somatic cell nuclei, and fresh, freeze-thaw (FT), and FD somatic cells treated with trehalose or epigallocatechin were compared. The length and morphology of the comet tail varied between each cell. Therefore, we measured whether the comet tail was present or absent. As shown in Fig. 2c, d and Supplementary Table 1, significantly more tails were seen in either FT or FD somatic cells than in fresh somatic cells. This result showed that not only FD somatic cells but also FT somatic cells had DNA damage, compared to fresh somatic cells.

We then injected FD somatic cell nuclei into enucleated oocytes (Fig. 3a–c). In the usual nuclear transfer method, injected nuclei form premature chromosome condensation (PCC) within 2 h, which is referred to as nuclear remodelling. However, because DNA damage in FD somatic cells is larger than that in fresh cells[22,23], we first examined the timing of PCC formation inside the reconstructed oocytes (Fig. 1 "2. PCC"). As shown in Fig. 4d to e and Supplementary Table 2, 1h after nuclear transfer, PCC formation was delayed in both FT and FD somatic cells compared to that in fresh somatic cells. When reconstructed oocytes were examined 2 h after nuclear transfer, the rate of PCC formation of FT and FD somatic cells treated with epigallocatechin increased and reached the same rate as that of fresh cells. However, the rate remained lower in FD somatic cells treated with trehalose, and the delay continued even 3 h after nuclear transfer (Fig. 3d, e).

When the reconstructed oocytes were activated and examined 6 h later, most embryos formed pseudo-pronuclei similar to the usual nuclear transfer (Fig. 4a). When cloned embryos derived from FD somatic cells were immunostained with anti-gamma-H2A.x antibody, which is a marker of DNA damage sites, numerous foci were detected (Supplementary Fig. 1 "3. γH2Ax", 4b,c, Supplementary Table 3). Given the difficulty in counting the number of foci inside the pseudo-pronuclei, we measured the brightness of the whole pseudo-pronucleus and then subtracted the brightness of the embryo cytoplasm. As shown in Fig. 4b, c, the pseudo-pronuclei derived from FD somatic cells were brighter than those from fresh or FT somatic cell samples. This result showed that the one-cell stage cloned embryos derived from FD somatic cells still possessed damaged DNA.

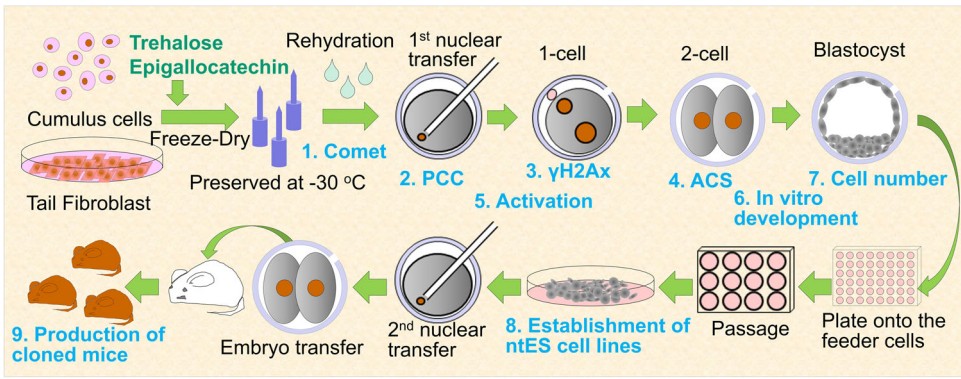

**Fig. 1 Schematic diagram of the present study.** Schematic showing the procedure for production of cloned mice from FD somatic cells. The FD somatic cells were preserved at –30 °C up to 9 months.

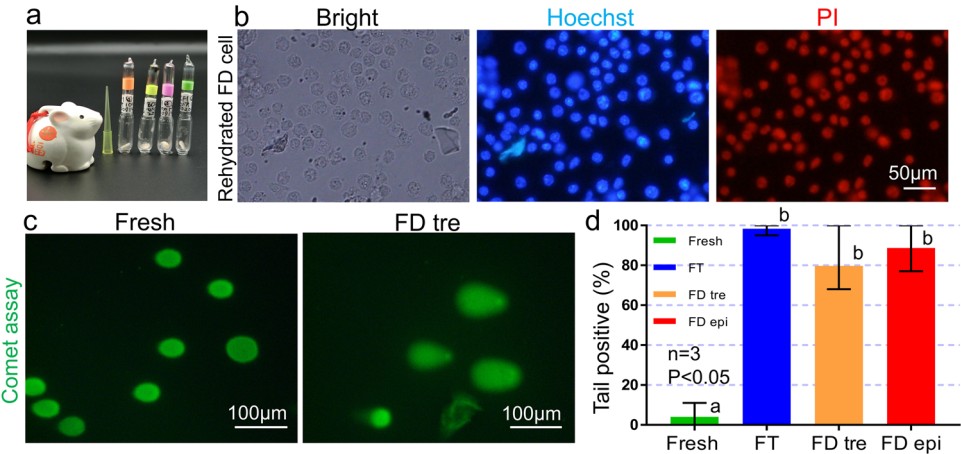

**Fig. 2 FD ampoules and DNA damage of FD somatic cell nuclei. a** Ampoules of FD somatic cells. **b** Damaged cell membrane after rehydration. Left: bright field; middle: Hoechst staining; right: propidium iodide (PI) staining. $n = 4$ biologically independent ampoules. **c**, **d** Comet DNA breakage assays of FD somatic cell nuclei. FT: Freeze-thaw. FD-tre: Freeze-dry with trehalose. FD-epi: Freeze-dry with epigallocatechin. $n = 3$ biologically independent experiment. **a** vs. **b**: $P < 0.05$ (one-way ANOVA and Tukey-multiple comparison-test). See Supplementary Table 1 for more details. Data are presented as mean value, minima and maxima.

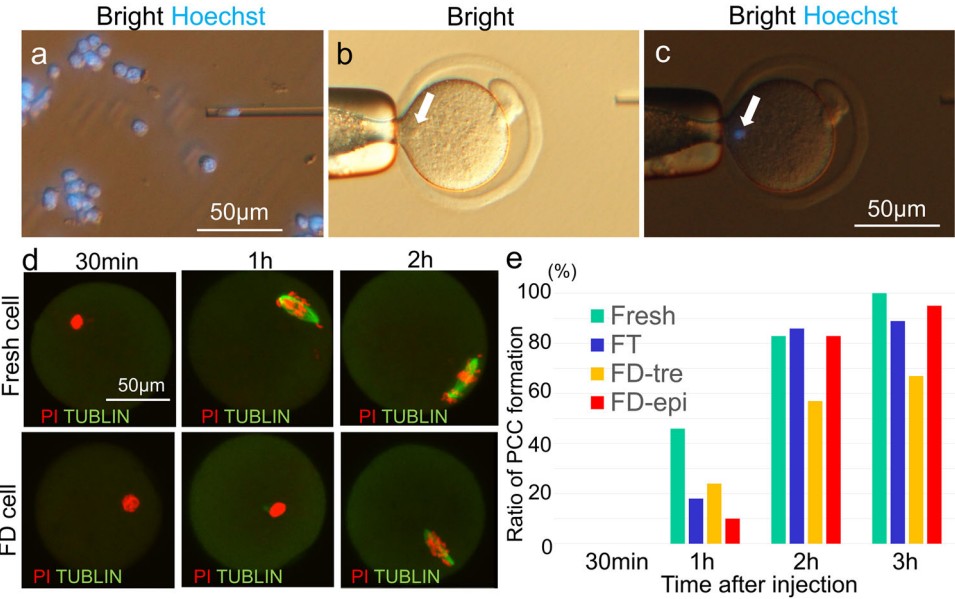

**Fig. 3 Nuclear injection and premature chromosome condensation (PCC). a–c** FD somatic cell nucleus was injected into enucleated oocytes; the arrow indicates injected nucleus. Somatic cell nuclei were stained by Hoechst33342. **d** Timing of PCC formation of fresh or FD somatic cells after nuclear injection into oocytes. **e** Percentage of PCC formation in oocytes from 30 min to 3 h after nuclear injection. $n = 18–40$ oocytes examined over 3 independent experiments. See Supplementary Table 2 for more details.

The incidence rate of abnormal chromosome segregation (ACS) at the 2-cell stage was determined as clustered and heavier DNA damage could not be detected by the previous assays[24]. Therefore, we examined the incidence rate of ACS at the 2-cell stage (Fig. 1 "4. ACS"). As shown in Fig. 4d, e and Supplementary Table 4, when fresh donors were used for nuclear transfer, 86% of cloned embryos showed normal chromosome segregation (NCS), but the rate was reduced to 53% by the FT treatment of the donor cells. The rate was further reduced to 14% when the donor cells were freeze-dried; this result suggests that freeze drying causes heavier DNA damage to the FD donor cell nuclei.

**Activation timing of reconstructed oocytes**. To determine the DNA repair capacity of the oocyte in the injected donor nucleus, we delayed the activation time of the reconstructed oocyte (Fig. 1

"5. Activation"). The in vitro developmental potential of the blastocysts was examined (Fig. 1 "6. In vitro development", Supplementary Table 5). When the reconstructed oocytes were activated at 30 min, they did not develop into blastocysts. However, when reconstructed oocytes were activated at 3 h or 5 h, PCC formation completed, as shown in Fig. 3e and we could obtain several cloned blastocysts from FD somatic cell nuclei (Fig. 5a, b). Although the blastocyst formation rate from FD donor cells was lower than that from fresh donor cells (Supplementary Table 5), the quality of the cloned blastocysts was similar between them (Fig. 1 "7. Cell number", 5c, d, Supplementary Table 6).

**Epigallocatechin in cloned embryo development in vitro**. We then examined the development rate of cloned embryos derived

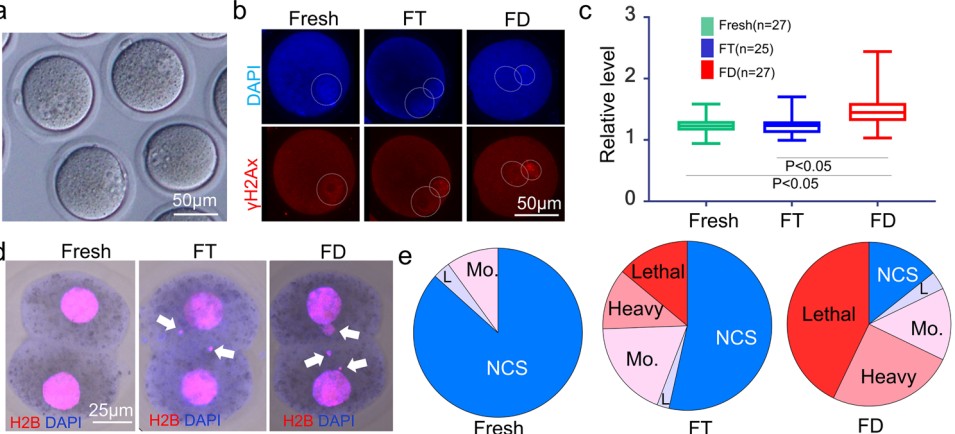

**Fig. 4 The foci of gamma-H2AX signals and abnormal chromosome segregation (ACS). a** Pseudo-pronuclear formation of cloned one-cell stage embryos derived from FD somatic cell nucleus. See Table 1, "No. of PN formed oocytes". **b** Immunostaining of zygotes derived from fresh control, FT somatic, and FD somatic cells using anti-gamma-H2AX antibodies. Nuclei are stained with 4',6-diamidino-2-phenylindole (DAPI; top, blue). The foci of gamma-H2AX signals show DNA double-strand breaks (bottom, red). **c** Plot of brightness of each pseudo-pronucleus. $n = 25–27$ 1-cell embryos examined three independent experiments. See Supplementary Table 3 for more details. Data are presented as mean value, minima and maxima. $P < 0.05$ (one-way ANOVA and Tukey-multiple comparison-test). **d** Detection of abnormal chromosome segregation (ACS) of cloned embryos derived from fresh, FT, and FD somatic cell nuclei. Arrows are judged as micronuclei using merged images. **e** Ratio of normal chromosome segregation (NCS: blue) and ACS (L: light; Mo: moderate; heavy and lethal) of cloned embryos derived from fresh, FT, and FD somatic cell nuclei.

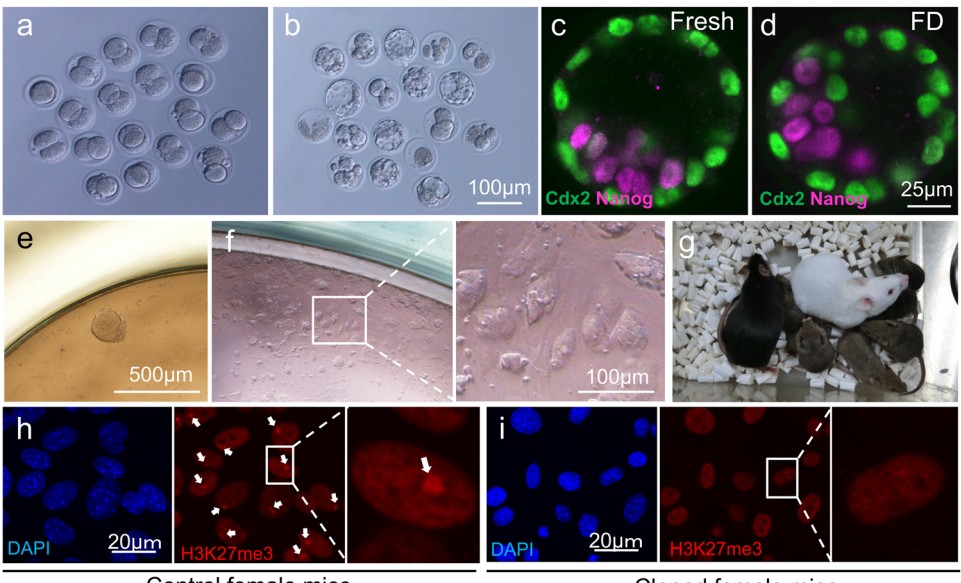

**Fig. 5 Establishment of ntES cell lines and production of cloned mice from FD somatic cells. a, b** Preimplantation development of cloned embryos at two-cell embryos and blastocysts derived from FD somatic cells. Those developed embryos were used for ntES cell establishment (see Table 1 for more details). **c, d** Immunostaining of cloned blastocysts derived from fresh (**c**) and FD somatic cells (**d**). The trophectoderm (CDX2-positve) show green. Inner cell mass (Nanog-positive) show red. $n = 3$ biologically independent experiment. See Supplementary Table 6 for more details. **e, f** Establishment of ntES cell lines from cloned blastocysts. Outgrowth observed 10 days after seeding in a 96-well plate (**e**). Three days after the first passage of this outgrowth (**f**). See Table 1 for more details. **g** The first cloned mouse "Dorami" derived from FD Cumulus cell nucleus. Dorami delivered the next generation by natural mating. **h, i** Immunostaining of tail-tip fibroblast of control (**h**) and cloned (**i**) female mice using anti-H3K27me3 antibodies. The nuclei of fibroblasts were detected using DAPI staining (blue). H3K27me3-positive spot (inactive X chromosome) is red (high magnification) and indicated by the arrow. Four cloned female mice and two control female mice were examined. See Supplementary Fig. 5.

from FD cumulus cells collected from several mouse strains (Table 1). Although cloned morulae were generated from all examined mouse strains, only one cloned blastocyst was obtained from the BDF1 strain (0.1%). However, when epigallocatechin was used to generate FD somatic cells, the rate of formation of cloned blastocysts derived from BDF1 cumulus cells was significantly improved (2.1%).

**Nuclear transfer ES (ntES) cell lines from FD somatic cells.** Although we succeeded in producing cloned blastocysts from FD somatic cells, the production rate was low. Similar to that in fresh cells, nuclear reprogramming was also difficult; the probability of obtaining cloned mice by transferring them into the uterus was very low. Therefore, we attempted to establish nuclear transfer embryonic stem (ntES) cell lines from cloned blastocysts (Fig. 1,

**Table 1 Establishment of ntES cell lines from freeze-dried male or female somatic cells treated with trehalose or epigallocatechin.**

| Donor cell type | Treated with | Mouse strain | No. of oocytes used | No. of activated oocytes | No. of PN formed oocytes | No. of embryo developed to | | | No. of ntES cell lines established | |
|---|---|---|---|---|---|---|---|---|---|---|
| | | | | | | 8-cell | Morula | Blastocyst | No. plated | No. established (/plate)[/PN] |
| ♀ FD Cumulus | Treha | BDF1 | 2153 | 1548 | 1229 | 85 (6.9) | 73 (5.9) | 1 (0.1)a | 150 | 15 (10.0) [1.2] |
| | | 129B6F1 | 520 | 381 | 200 | 16 (8.0) | 12 (6.0) | 0 | 28 | 5 (17.9) [2.5] |
| | | BD129F1 | 337 | 304 | 298 | 14 (4.7) | 2 (0.7) | 0 | 16 | 2 (12.5) [0.7] |
| | | C3H/He | 407 | 270 | 205 | 3 (1.5) | 8 (3.9) | 0 | 11 | 0 |
| | | C57BL/6 | 130 | 104 | 84 | 2 (2.4) | 1 (1.2) | 0 | 3 | 0 |
| ♀ FD Cumulus | Epigallo | BDF1 | 1994 | 1779 | 1425 | 77 (5.4) | 63 (4.4) | 30 (2.1)b | 119 | 16 (13.4) [1.1] |
| ♂ FD Fibroblast | Epigallo | BDF1 | 931 | 735 | 453 | 31 (6.8) | 18 (4.0) | 8 (1.8)b | 41 | 4 (9.8) [0.9] |
| | | BCF1 | 597 | 520 | 277 | 31 (11.2) | 35 (12.6) | 0 | 65 | 4 (6.2) [1.4] |
| ♀ FD Fibroblast | Epigallo | BDF1 | 384 | 308 | 167 | 4 (2.4) | 5 (3.0) | 2 (1.2)b | 11 | 3 (27.3) [1.8] |
| | | BCF1 | 697 | 574 | 410 | 18 (4.4) | 18 (4.4) | 0 | 35 | 2 (5.7) [0.5] |
| | | BD129F1 | 192 | 158 | 54 | 4 (7.4) | 0 | 0 | 4 | 1 (25.0) [1.9] |
| Control ♀Fresh cumulus | - | BDF1 | 132 | 126 | 103 | 11 (10.7) | 64 (62.1) | 11 (10.7) | 47c | 25 (53.2) [24.3] |
| ♀ FT cumulus | - | BDF1 | 93 | 87 | 82 | 21 (25.6) | 11 (13.4) | 1 (1.2) | 12 | 5 (41.7) [6.1] |
| ♀ Fresh fibroblast | - | BDF1 | 260 | 188 | 162 | 28 (17.3) | 49 (30.2) | 3 (1.9) | 72 | 6 (8.3) [3.7] |

*Treha* Trehalose, *Epigallo* Epigallocatechin.
a vs. b: $P < 0.01$; $\chi^2$-test. Compared between FD groups.
cSome of embryos were not plated to reduce the costs.

"8. Establishment of ntES cell lines"). The cloned blastocysts were likely derived from less damaged donor FD somatic cells, and the resulting ntES cells were also less damaged; we could use these ntES cells as better quality and easily prepared donor cells for serial nuclear transfer.

We successfully established ntES cell lines from blastocysts derived from FD cumulus cells treated with trehalose of F1 mouse strains, with 10–18% success rate (Fig. 5e, f; Table 1 and Supplementary Table 7). These FD cumulus cells were stored at −30 °C for 1 week to 9 months before use. However, we were unable to establish any ntES cell lines from inbred donor strains. When epigallocatechin was used instead of trehalose to prepare FD somatic cells, although developmental rate to blastocysts were high, the establishment rate of ntES cell lines (13.4%) was the same as when trehalose was used. Although the establishment rate of ntES cell lines from reconstructed oocytes was lower than that of fresh cells as donors, all established ntES cell lines showed normal morphology (Fig. 5f). All the examined ntES cell lines had a normal karyotype (38–68%, Supplementary Table 8).

Next, we tried to generate FD fibroblasts from the male and female tail tips. The developmental rate to the cloned morulae/blastocyst using FD fibroblasts was similar to that of FD cumulus cells, regardless of their sex or mouse strain (Table 1). Using these cloned embryos, we were able to establish several ntES cell lines from all strains or sexes, at 6–27% success rate, when stored at −30 °C for 2 weeks to 8 months (Table 1 and Supplementary Table 7). Most of the examined ntES cell lines had a normal karyotype (50–76%), except for two lines (# FDF3 and # FDF5; Supplementary Table 8).

**Cloned mice from FD somatic cells via ntES cell lines.** Finally, we attempted to produce cloned mice from ntES cell nuclei as donors for second-round nuclear transfer (Fig. 1 "9. Production of cloned mice"); we succeeded in producing 75 cloned mice (Table 2). All cloned mice possessed hypertrophic placenta (Supplementary Table 9), which is a typical abnormality in cloned mice[25,26], but the body weights were within the normal range for somatic cell cloned mice[25]. BCF1 male cloned mice showed agouti coat colour (Supplementary Fig. 3), which demonstrates the origin of these donor cells because oocytes were collected from the BDF1 strain (black coat colour) and recipient females were used as ICR strains (white coat colour). The first cloned mouse, "Dorami", was derived from FD cumulus cells and grew into adulthood. After examination of her fertility, she was kept alone to examine longevity and died 676 days (1.9 years) later.

**Fertility of cloned mice.** The fertility of the cloned mice is an important factor in the viability of this technique for preservation of genetic resources. After maturation, we randomly selected nine female and three male cloned mice, which were mated with male and female natural mice, respectively. After 2–3 months, all females delivered the next generation of mice (Fig. 5g and Supplementary Table 10). These results clearly suggest that the cloned mice derived from FD somatic cells possess normal fertility. In addition, the testes or ovaries of cloned mice were histologically examined to examine the status of the reproductive organs, and compared with control mice, no obvious abnormalities were observed (Supplementary Fig. 4).

**Female-cloned mice derived from male FD somatic cells.** Interestingly, all cloned mice generated from the FDF3 ntES cell line derived from male FD fibroblasts were born as females. We confirmed the localisation of trimethylated histone H3 at lysine 27 (H3K27me3) in fibroblasts of cloned mice, which are responsible for the repressive chromatin state in inactive X[27]. The

**Table 2 Production of cloned mice from freeze-dried somatic cell via ntES cells.**

| Donor cell type | Sex | Treated with | Mouse strain | No. of oocytes used | No. of activated oocytes | No. of PN formed oocytes | No. of embryos developed to 2-cell stage (/PN) | No. of embryo transfer | No. of offspring (/ET)[/PN] |
|---|---|---|---|---|---|---|---|---|---|
| Cumulus | F | Treha | BDF1 | 3146 | 2695 (85.7) | 2356 (74.9) | 1400 (59.4) | 1327 | 3 (0.2) [0.1] |
| | | Epigallo | BDF1 | 2369 | 2116 (89.3) | 1875 (79.1) | 1016 (54.2) | 1016 | 16 (1.6) [0.9] |
| Fibroblast | M | Epigallo | BDF1 | 1868 | 1647 (88.2) | 1460 (78.2) | 867 (59.4) | 864 | 47[a] (5.4) [3.2] |
| | | Epigallo | BCF1 | 1200 | 1012 (84.3) | 884 (73.7) | 520 (58.8) | 520 | 9 (1.7) [1.0] |

*Treha* Trehalose, *Epigallo* Epigallocatechin.
[a]All cloned mice derived from a donor XO ntES cell line became female.

control female fibroblasts showed one localisation on the inactivated X chromosome (Fig. 5h), whereas female-cloned mice did not display this localisation signal in any cells (Fig. 5i and Supplementary Fig. 5). In addition, karyotype analysis showed that the most frequent number of chromosomes was 39 in this ntES cell line and in cloned mice (58–65%, Supplementary Fig. 6, Supplementary Table 8, 11). These results suggest that the Y chromosomes were lost in these cloned mice. However, as mentioned above, female-cloned mice derived from a male mouse were able to successfully deliver pups after mating with a natural male (Supplementary Fig. 7). These results suggest that even if Y chromosome loss does occur, this technique can still be used to the available genetic resources in extreme circumstances, such as almost extinct species.

## Discussion

Oocytes have a strong DNA repair capability, and the DNA repair factor in oocyte cytoplasm is quite abundant[23,28]. Several groups, including ours, have produced cloned blastocysts from FD somatic cells by nuclear transfer. However, cloning of animals from cloned blastocysts has not yet been successful[22,29–32]. This could be because cloned embryos were generated from somatic cell nuclei with incomplete DNA damage repair, which is caused by freeze drying. This theory is confirmed using extensive DNA damage and ensuing repair activity in FD somatic cells in cloned embryos[22,23]. In this study, we found that many cloned embryos derived from FD somatic cells showed DNA damage at the one-cell stage or chromosomal abnormalities at the 2-cell stage (Fig. 4d, e). In contrast, epigallocatechin, which has an anti-oxidant effect on spermatozoa[33] and somatic cells[34], improved the nuclear remodelling rate and the developmental rate of blastocysts when added to the medium before FD treatment (Table 1); however, the DNA damage did not improve. This result may suggest that epigallocatechin could protect cytoplasmic factors, such as histones, rather than DNA.

Another possible reason animals have not been cloned from FD somatic cells is incomplete nuclear reprogramming of donor FD somatic cells. In the present study, PCC formation or nuclear remodelling was delayed when FD somatic cell nuclei were injected into oocytes, compared to fresh or freeze-thawed somatic cell nuclei (Fig. 3d, e). This result suggests that even if the oocyte has abundant DNA repair factors[23], it requires a long period to repair severely damaged DNA of FD somatic cells; subsequent nuclear remodelling is also delayed. Although the relationship between nuclear remodelling and reprogramming remains unknown, if they occur consecutively, activating a reconstructed oocyte before complete remodelling may initiate cloned embryo development with incomplete reprogramming of the donor nucleus. Such embryos cannot develop fully. However, if excessive time is devoted to DNA repair of the donor nucleus, the developmental potential of the reconstructed oocyte will be reduced due to ageing[35], even if the donor nucleus is sufficiently

reprogrammed. Determining the optimal activation time for reconstructed oocytes is important for the production of cloned animals from FD somatic cells.

We attempted to establish ntES cell lines from cloned blastocysts without transferring them into recipient females (Fig. 1), because the establishment rate of ntES cells from cloned embryos is much higher than the birth rate of cloned mice[21,36,37]. We succeeded in establishing ntES cell lines from either female or male tail-tip fibroblasts or female cumulus FD cells. The cloned mice can be produced from the ntES cell line through the serial nuclear transfer of female or male FD somatic cell nuclei. Unfortunately, because the establishment rate of ntES cell lines is ~1% (Supplementary Table 1) and the birth rate of cloned mice from those ntES cell nuclei is ~2% (Table 2), the total success rate of cloned mice from FD somatic cells is only 0.02%. This success rate is much lower than that in the world's first cloned animal, the sheep Dolly (0.4%)[17]. Furthermore, female-cloned mice were produced from a male donor mouse, which indicated that this set of techniques may result in chromosome loss in cloned offspring. If the objective is to produce completely identical clones from the donor, these female-cloned mice from a male donor would not be considered a success, which would further reduce the ultimate success rate. However, if sex-changed cloned mice are included in the tally, a total of 75 cloned mice were born from six out of 49 ntES cell lines. When calculated from only the successful cell lines, the success rates of cloned mice were 1–7% (Supplementary Table 12), which were comparable to the birth rate of cloned mice from ES/ntES cells[18,36]. This result is very common in the ntES cell line; when some ntES cell lines were established from the same individual, each line had different features and different success rates of cloned mice[38].

The first successful offspring from FD spermatozoa was produced more than 20 years ago[4], but the success of FD treatment in cells other than spermatozoa has not been reported[22,29,31,32]. Sperm nuclei are tightly condensed by protamine, whereas somatic cells have a more fragile nuclear structure, a high water content, and numerous histones compared to spermatozoa[39,40]. The tolerance of spermatozoa to FD damage is considered to be due to their specific nuclear structure. Additionally, spermatozoa are haploid; therefore, only half the amount of DNA was damaged by FD treatment compared to diploid somatic cells.

Although the current birth rate of cloned mice from FD somatic cells remains low and the cells must be preserved at −30 °C, somatic cells can be stored in the FD state for a long period and used as a donor for producing offspring. Somatic cells can be easily collected from any animal, including infertile or immature/aged animals; hence, somatic cells can become a valuable method of preserving genetic resources once they are stored at ambient temperature.

It should, however, be noted that current cloning techniques are not perfect, and it is likely that cloned offspring will have some epigenetic abnormalities due to incomplete reprogramming[41].

Nevertheless, we have previously demonstrated that the next generation of cloned mice is normal even after nuclear transfer was repeated 25 times[42]. In this study, although cloned mice were produced by repeating nuclear transfer twice, these clones were fertile and therefore did not detract from the objective of preserving genetic resources.

Interestingly, one of the ntES cell lines derived from male FD fibroblasts lost its Y chromosome and became an XO cell line; all the cloned mice produced from that cell line became female. Sex change occurs rarely in cloned animals[43], and Y chromosome deletion has been reported during ES cell line culture[44]. In this study, we did not determine the precise point at which the Y chromosome was lost, such as during FD treatment, nuclear transplantation, or the establishment of ntES cells, and this result does represent a failure to completely preserve the genetic resources of the donor mouse. However, if the same treatment could be performed in endangered species where only males survived, it would be possible to produce females and naturally preserve the species. Ultimately, the preservation of somatic cells by FD treatment will be an important method supporting the establishment of alternative, cheaper, and safer bio-banking solutions.

## Methods

**Animals**. BDF1 (C57BL/6N × DBA/2), BCF1 (C57BL/6N × C3H/He), 129B6F1-GFPTg (129/Sv × C57BL/6N-GFP), BD129F1 (BDF1 × 129/Sv), C3H/He, C57BL/6N, and ICR mice (8–10 weeks of age) were obtained from SLC Inc. (Hamamatsu, Japan) or produced in our mouse facility. Surrogate pseudo-pregnant ICR females, which were used as embryo recipients, were mated with vasectomised ICR males, the sterility of which had been demonstrated previously. On the day of the experiment or after finishing all experiments, the mice were euthanised by $CO_2$ inhalation or cervical dislocation. All animal experiments were conducted in accordance with the Guide for the Care and Use of Laboratory Animals and were approved by the Institutional Committee of Laboratory Animal Experimentation of the University of Yamanashi (reference number: A29–24), which followed the ARRIVE guidelines.

**Collection of oocytes and cumulus cells**. Female mice were super-ovulated through the injection of 5 IU of equine chorionic gonadotropin, followed by 5 IU of human chorionic gonadotropin (hCG) after 48 h. Cumulus–oocyte complexes (COCs) were collected from the oviducts of females 14–16 h later and moved to a Falcon dish containing HEPES–CZB medium[45]. To disperse the cumulus cells, COCs were transferred to a 50 μL droplet of HEPES–CZB medium containing 0.1% bovine testicular hyaluronidase for 3 min. Cumulus-free oocytes were washed twice and transferred to 20 μL droplets of CZB medium[46] for culturing. Simultaneously, the remaining cumulus cells were collected and used for FD treatment or introduced into PVP medium[45] on the manipulation chamber for fresh control.

**Collection of tail-tip fibroblasts**. To collect tail-tip fibroblast cells, tail tips were freed from the skin, cut into small pieces, and incubated in 5 mL DMEM (Sigma-Aldrich, St. Louis, MO, USA) supplemented with 10% foetal calf serum (Sigma-Aldrich). After 10–14 days at 37.5 °C under 5% $CO_2$ in air, proliferating fibroblasts were dissociated using trypsin and replated into a larger dish to increase the cell number. This process was repeated twice. Then, some of them were frozen at −80 °C, and others were used for nuclear transfer as fresh or FD somatic cell experiments.

**Preparation of FD somatic cells**. Both cumulus cells and fibroblasts were suspended in Tris–EGTA medium[47,48] with trehalose or epigallocatechin (Theliokeep, Bio Verde, Kyoto, Japan). Aliquots (100 μL) of the cell suspension were dispensed into glass ampoules. The ampoules were stored at 4 °C for 3 h, −30 °C for 3 h, and −80 °C for 6–24 h until use. For the FD treatment, frozen ampoules were placed in LN2 for at least 5 min and then freeze-dried using an FDU-2200 freeze dryer (EYELA, Tokyo, Japan). The cork of the freeze dryer was open for at least 3 h until all the samples were completely dry. After drying, the ampoules were sealed by melting the ampoule necks using a gas burner under vacuum, as described previously[9]. Those ampoules were preserved at −30 °C for at least two days to up to 9 months until use.

**Rehydration of FD somatic cells and observation**. Immediately before the nuclear transfer, the neck of the glass ampoule was broken, and 100 μL of sterile distilled water was immediately added and mixed using a pipette. Then, 2 μL of the suspension was placed in the manipulation chamber for observing the FD somatic cells. Using microcapillaries, FD somatic cells were collected and moved to different drops in the same chamber to observe the morphology of the rehydrated cells. Some of them were mixed with Hoechst 33342 and PI and then observed under UV light to determine whether they had intact cell membranes.

**Analysis and scoring of comet slides**. The single-cell gel electrophoresis technique (designated as the comet assay) measures DNA damage, including double- and single-strand breaks[49]. Thus, comet assays were used to detect DNA damage of FD cumulus cell nuclei, according to the manufacturer's protocol (Trevigen, MD, USA). Briefly, specimens were collected from ampoules immediately after opening and rehydrated in water and then mounted on four slides, and 100–300 cells were analysed using electrophoresis. In this study, the length and morphology of comet tail varied between cells. Therefore, we measured whether the comet tail was present or absent rather than measuring the tail lengths.

**Nuclear transfer**. Oocytes were transferred to a droplet (~10 μL) of HEPES–CZB, containing 5 μg/mL cytochalasin B that was placed under oil in the operating chamber of a microscope stage. The oocyte was held with an oocyte-holding pipette, and its zona pellucida was perforated by applying several piezoelectric pulses with the tip of an enucleation pipette[18]. The metaphase II chromosome spindle complex, distinguished as a translucent spot in the ooplasm, was drawn into the pipette with a small amount of accompanying ooplasm and gently pulled away from the oocyte until the stretched cytoplasmic bridge was pinched off.

A donor cell was drawn in and out of the injection pipette until the plasma membrane was disrupted. Occasionally, a few piezoelectric pulses were applied to break the membranes. Donor nuclei were injected into enucleated oocytes as described previously[18]. Briefly, the zona pellucida of the enucleated oocyte was perforated by applying several piezoelectric pulses, and then, the membrane of the oocyte was perforated with a few more piezoelectric pulses. The donor nucleus was then inserted into the ooplasm. After the donor nucleus was injected, the reconstructed oocytes were transferred into a droplet of KSOM medium and incubated at 37 °C under 5% $CO_2$.

**Activation of cloned embryo**. The reconstructed oocytes were activated using 5 mM $SrCl_2$ in $Ca^{2+}$-free CZB medium in the presence of 50 nM trichostatin A (TSA), supplemented with 5 μM latrunculin A for 9 h[50–52]. Pseudo-pronuclear formation was examined, and cloned embryos were cultured for 96 h to examine their potential to develop into blastocysts and were then used for establishing ntES cell lines.

**Immunostaining of reconstructed oocytes and zygotes**. Thirty minutes to three hours after nuclear injection, or 10 h after the activation treatment, the oocytes were fixed for 20 min at 25 °C in 4% (w/v) paraformaldehyde. The fixed oocytes were washed thrice in PBS–polyvinyl alcohol (0.1 mg/mL PVA, Sigma-Aldrich, St Louis, MO, USA) for 10 min and stored overnight at 4 °C in PBS, supplemented with 1% (w/v) bovine serum albumin (BSA, Sigma-Aldrich) and 0.1% (v/v) Triton X-100 (Nacalai Tesque, Inc., Kyoto, Japan). For observing remodelled donor nuclei in reconstructed oocytes, the primary antibodies used were an anti-beta-tubulin mouse monoclonal antibody labelled with FITC (1:1000; Sigma-Aldrich F2043) and cultured in 0.1% Triton X-1% BSA/PBS for 2 h at ambient temperature. After each oocyte had been washed thrice in PBS–PVA for 10 min, DNA was stained with PI (Sigma-Aldrich). The nuclear remodelling of injected donor nuclei was categorised into three groups: "Intact", "Early PCC", and "PCC". Early PCC was confirmed when β-tubulin was observed around the nucleus but did not form PCC. PCC was determined when the PCC and spindle were detected.

For the observation of DNA damage in cloned zygotes, the primary antibodies used were anti-phospho-H2AX (Ser139) rabbit polyclonal antibody (1:500; Millipore-Merck, Darmstadt, Germany, 07-164) and an anti-histone H3 (dimethyl K9) mouse monoclonal antibody (1:500, Abcam, Cambridge, UK, ab1220). The secondary antibodies used were Alexa Fluor 488-labelled goat anti-mouse IgG (1:500, Molecular Probes, Eugene, OR, USA) and Alexa Fluor 568-labelled goat anti-rabbit IgG (1:500 dilution; Molecular Probes). DNA was stained with 4′6-diamidino-2-phenylindole (DAPI; 2 μg/mL; Molecular Probes). The brightness of the whole male pronucleus was measured using ImageJ software and was then subtracted from the brightness of the zygote cytoplasm.

**Detection of abnormal chromosome segregation**. Six hours after the activation treatment on the reconstructed oocytes, the cloned zygotes were injected with histone H2B-mCherry mRNA for visualising their nuclei or micronuclei. The next day, 2-cell stage embryos were fixed and permeabilised with 4% PFA and 0.5% Triton X-100 for 15 min. These embryos were observed using confocal fluorescence microscopy (Olympus FV1200, Tokyo, Japan) in DAPI and 1% BSA-containing PBS. To reduce nonspecific misidentification, we only used mCherry and DAPI double-positive signals in the analysis. ACS was categorised into four groups: "light", "moderate", "heavy", and "lethal" (Supplementary Fig. 4, upper picture). Light ACS was judged when only one micronucleus was detected (Supplementary Fig. 4 lower table). Moderate ACS was judged when two small, one to two medium, or one large micronucleus was detected. Heavy ACS was judged when three small, medium, or two or three large micronuclei were detected. Lethal ACS was judged

when the embryos had multiple micronuclei. Importantly, when two conditions co-occurred, the evaluation became more severe. For instance, when one moderate and two small micronuclei were observed in the embryo, it was considered as "heavy".

**Immunostaining of blastocysts**. To evaluate the quality of the cloned blastocysts from FD somatic cells, cell numbers were examined using immunofluorescence staining as previously described[12]. The primary antibodies used were an anti-CDX2 mouse monoclonal antibody (1:500; BioGenex, San Ramon, CA, USA, MU392A-UC) for detecting the TE cells and an anti-Nanog rabbit polyclonal antibody (1:500; Abcam, Cambridge, UK, ab80892) for detecting the ICM cells. The secondary antibodies used were Alexa Fluor 488-labelled goat anti-mouse IgG (1:500; Molecular Probes Inc., Oregon, USA, A11029) and Alexa Fluor 568 goat anti-rabbit IgG (1:500; Abcam, Cambridge, UK, A11036). DNA was stained with DAPI (2 µg/mL; Molecular Probes).

**Establishment of ntES cell lines**. When cloned embryos developed to the morula/blastocyst stage, they were treated with acid Tyrode solution to remove the zonae pellucidae and used to establish ntES cell lines as described previously[36] with a slight modification. Embryos were placed in 96-multi-well dishes precoated with mouse embryonic fibroblasts in 20% Knock-out Serum Replacement (Invitrogen, Company, CA) and 0.1 mg/mL ACTH (fragments 1–24; American Peptide Company, Sunnyvale, CA, USA). Proliferating outgrowths were dissociated using trypsin and replated to fibroblasts until stable cell lines grew out. The established ntES cell lines were also used as donor cells for nuclear transfer.

**Karyotyping of cells and detection of inactive X chromosome**. To increase the metaphase stage of ntES cells or fibroblasts, 10 µL/mL of demecolcine (Wako, Japan, 045-18761) was added to the medium and cultured for 2 h. Those cells were detached by trypsin, exposed to 0.075 M KCl solution for 20 min and fixed with Carnoy's solution, and then applied onto the glass slides. To count the number of chromosomes, the glass slides were stained with Giemsa or DAPI and observed under a microscope. To detect inactive X chromosomes, fibroblasts were cultured on the glass, fixed, and observed after immunofluorescence staining. Primarily, we used anti-H3K27me3 rabbit polyclonal antibodies (1:500; Millipore-Merck, Darmstadt, Germany 07-449). The secondary antibodies used were Alexa Fluor 568 goat anti-rabbit IgG (1:500; Abcam, Cambridge, UK, A11036). DNA was stained with DAPI (2 µg/mL; Molecular Probes).

**Production of cloned offspring using ntES cell nuclei**. Enucleated B6D2F1 oocytes were injected individually with the ntES cell nuclei derived from FD cumulus cells or FD fibroblasts and then activated as described above. On the day after the night when the mice had been mated with a vasectomised male, cloned embryos that had reached the 2-cell stage were transferred into the oviducts of pseudo-pregnant ICR female mice at 0.5 days post coitum (dpc) before embryo transfer. On the day of embryo transfer, the recipients were anaesthetised using an intraperitoneal injection of medetomidine, midazolam, and butorphanol. Five to eight embryos were transferred into each uterine horn, and an equal amount of atipamezole was injected. At 19.5 dpc, the offspring were delivered by caesarean section. Randomly selected offspring were transferred to the cage of a foster mother who had delivered pups naturally. Three weeks later, the offspring were mated, and their fertility was examined.

**Histological analysis**. The recovered testis or ovary from cloned mice and control mice was fixed in 4% paraformaldehyde in PBS for 48 h and embedded in OCT compound (Sakura Finetek, Japan). The OCT blocks were sectioned at a thickness of 10 µm with a cryostat (Thermo Fisher Scientific, USA) and mounted on Superfrost Micro Slides (Matsunami Glass, Japan). Serial cross sections were stained with haematoxylin and eosin for light microscopy.

**Statistical analysis**. All experiments were repeated at least three times; these studies were performed independently by two to three, and similar results were obtained irrespective of the experimentalists. The rates of embryo development, birth of offspring, and the body and placental weights were evaluated using chi-squared tests. Fluorescence levels were evaluated using a one-way ANOVA and Tukey-multiple comparison-test (Prism, GraphPad Software, USA), and $P < 0.05$ was considered to represent a statistically significant difference.

**Reporting summary**. Further information on research design is available in the Nature Research Reporting Summary linked to this article.

## Data availability
All data generated or analysed during this study are included in this published article. Source data are provided with this paper.

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

## Acknowledgements

We thank Dr. T. Kohda, Dr. S. Kishigami, Dr. M. Ooga, Dr. Y. Fujimoto, Mr. M. Nakamura, Mr. M. Sakamoto, Mr. Y. Sato, Mrs. C. Yamaguchi, and Y. Kanda for assistance in preparing this manuscript. This work was partially funded by Research Fellowships from the Japan Society for the Promotion of Science for Young Scientists (JP20J23364) to D.I.; grants from MEXT Grant-in-Aid for Scientific Research on Innovative Areas (JP19H05756) to T.I.; the Naito Foundation and Takahashi Industrial and Economic Research Foundation (189) to S.W.; and from the Asada Science Foundation and the Canon Foundation (M20-0008) to T.W. The authors would like to thank Editage for English language editing.

## Author contributions

S.W. and T.W. conceived and designed the study. S.W., D.I., E.H., T.I., and T.W. performed experiments, analysed the data, and interpreted the results. S.W. and T.W. wrote the manuscript. All authors read and edited the manuscript.

## Competing interests

The authors declare no competing interests.
