## [Peer Review File · Nature Communications]

Healthy cloned offspring derived from freeze-dried somatic cellsREVIEWER COMMENTS

Reviewer #1 (Remarks to the Author):

In this manuscript, the authors described the success to generate cloned mouse embryos and offsprings from freeze-dried mouse somatic cells. While the experiments were designed properly and data solid, there are several concerns about the novelty and the potential application of such technology.

1. The feasibility to obtain cloned embryos and offsprings from freeze-dried germ cells of various species have been published a long time ago, the conceptual advance of this study is limited.
2. The key challenge of cloned individuals is the genomic instability. The freeze-dried treatment apparently increases genomic instability dramatically. While the authors suggest that the genomic instability of the offsprings recloned from the cloned embryos is reduced, they failed to provide data on the nature and extent of the genomic instability of the recloned individuals, raising doubt to use this technology to preserve genetic resources.
3. To support the conclusion that this cloning technology has no negative influence on development, the authors need to provide more extensive data on the fertility and normal status of various organs in the cloned mice.

Reviewer #2 (Remarks to the Author):

Remark for Authors

Review of MS: Healthy cloned offspring derived from freeze-dried somatic cells, by Sayaka Wakayama et al.

The manuscript reports the production, for the first time, of normal and fertile mice following the use of ES cells established from blastocyst cloned using freeze dried somatic cells as a donor. Intermediate steps leading to this achievement, described and discussed in the manuscript, were the characterization of the damage exerted by the freeze dried procedure, both at whole cells and DNA level, including the formation of micronuclei, expression of bold genomic damage, in embryos derived from freeze dried cells. Each step configured itself as a limiting factor, but at the end, through the correction measures adopted by the authors, embryos and embryonic stem cells were derived from dry mouse granulosa and fibroblast cells.

The data are robust, and backed up by thousands of nuclear transfer embryos, and hundreds of ES lines. The demonstration that dry nuclei can develop into normal adults is a remarkable achievement, both for the scientific community and lay audience. Indeed, the presented work is a significant step toward the establishment of alternative, cheaper and safer bio-banking solutions.

Minor Points:

Page 2, line 23: ...In this study, we aimed to generate a cloned mice from freeze-dried (FD) somatic cell nuclei, the cells were preserved at -30 °C for up to 9 months. The sentence is not clear: were the cells stored for 9 month at -30 °C prior drying? Please clarify

Page 2, line 25: and nucleic DNA damage significantly increased when cells were rehydrated. As presented, seem that DNA damage is due to the rehydration, rather that after freeze drying. Please, emend.

Page 2, line 27: Using these cells as nuclear donors for re-cloning, we obtained healthy cloned mice from FD somatic cell nuclei in male or female mice. Better if: Using these cells as nuclear donors for

re-cloning, we obtained healthy cloned female and male mice. Please change.

Page 2, line 32: FD somatic cells will be a superior and safer source. Superior here is inappropriate, fresh cells are "superior" in all respect. Please, change the text accordingly.

Page 3, line 60: improving the nuclear transfer procedure. This is not correct, simply, the procedure has been adapted to the kind of nuclei used. Please, change the text accordingly.

Page 7, line 142: ...we could use these ntES cells as infinite donors for serial nuclear transfer. The wording here is not appropriated, please modify the text.

Page 11, line 240: whereas somatic cell nuclei are not hard,.....better : somatic cells have a more fragile nuclear structure, comparing to spermatozoa.

Page 11, line 245: However, this study clearly demonstrated that neither protamine nor half the amount of DNA are essential for successful cloning. The sentence does not add valid elements to the discussion, please remove.

Page 12, line 260: ... will be an important method for the permanent prosperity of humanity. Reads pleonastic. Better to remain within the possible application of the outputs achieved, like the establishment of biobank at low cost, for any purpose, biodiversity, bio medicine, etc.

Reviewer #3 (Remarks to the Author):

Wakayama is an acknowledged expert in the field of mammalian transfer and what he has done here is, I think, valuable. I think that it is essential that the paper is reformatted so that the summary makes very clear the points of the work. That is that the summary should say: what percent of normal adult nuclear transfer was achieved. It should also put the main points of the work in a clear abstract.

Additionally, I asked a graduate student of mine to review this paper, and below you can find his comments about this paper which I agree with.

My graduate student's opinions: We suggest the editors accept the paper. It is remarkable that the authors reach around 14% efficiency, especially in experiments as delicate as nuclear transfer. It is important that the authors mention the downside of dry-freezing and the significant damages observed in the DNA. We believe the authors should make a specific emphasis on how toxic this procedure is and exclude as successful nuclear transfer the specific cases where complete chromosomes were degraded and/or lost. Independently of whether the nuclear transfer developed into adults, loss of complete chromosome is just another proof of the detrimental effects of dry-freezing. We believe the total number of success cases should not include those where sexual chromosomes were lost.

Importantly, we also observed a substantial amount of self-citations, ignoring important work in the field such as Wilmut et al. 1997 and Gurdon et al. 1958/1962. We recommend that after doing these minor alterations the paper should be published in this journal.

Point-by-point response to the reviewers' comments:

Reviewer #1 (Remarks to the Author):

In this manuscript, the authors described the success to generate cloned mouse embryos and offsprings from freeze-dried mouse somatic cells. While the experiments were designed properly and data solid, there are several concerns about the novelty and the potential application of such technology.

We thank you for your important comments and advice.

Our study demonstrates for the first time that offspring can be generated from FD somatic cells, and we believe that this technology is particularly valuable for preserving genetic resources.

1. The feasibility to obtain cloned embryos and offsprings from freeze-dried germ cells of various species have been published a long time ago, the conceptual advance of this study is limited.

Since our first success in the production of offspring from FD sperm in 1998¹, there have been many successful studies that have employed this technique in many animal species. However, until the present study, there has been no success in producing offspring from germ cells (oocytes or embryos) other than spermatozoa. Moreover, there are no examples of somatic cells successfully producing offspring after FD treatment. In the case of FD somatic cells, the production of offspring requires not only an improved FD method but also a combination of cloning techniques, which was thought to be much more difficult than producing offspring from FD sperm, and indeed, no one has proven successful in this endeavour so far. Therefore, the successful production of offspring from FD somatic cells that we report in this manuscript is completely novel and represents substantial progress in this field.

The technique of producing offspring from FD somatic cells is of great value for the preservation of genetic resources. Spermatozoa can be obtained in large numbers from healthy males, but only a few oocytes or embryos can be obtained even from healthy females. In addition, it is impossible to obtain spermatozoa as well as oocytes/embryos from infertile, young, or old individuals. On the other hand, somatic cells, unlike germ cells, can be obtained in large quantities from any individual and from anywhere in the body. At present, the success rate of cloned offspring from FD somatic cells is too low to be used for the preservation of genetic resources, but with future improvements, we believe that this technique has the potential to become more valuable than the storage of germ cells.

2. The key challenge of cloned individuals is the genomic instability. The freeze-dried treatment apparently increases genomic instability dramatically. While the authors suggest that the genomic instability of the offsprings re-cloned from the cloned embryos is reduced, they failed to provide data on the nature and extent of the genomic instability of the re-cloned individuals, raising doubt to use this technology to preserve genetic resources.

As you have pointed out, FD treated cells have been shown to exhibit significant DNA damage. This is probably physical damage caused by the FD process (e.g., crystallization of medium components in the nucleus). However, the abnormalities seen in the cloned mice may also be due to incomplete reprogramming by the oocyte cytoplasm and are thus considered epigenetic abnormalities rather than DNA abnormalities. It has already been demonstrated that epigenetic abnormalities produced by cloning techniques (nuclear reprogramming) are not transmitted to the next generation ². In this study, nuclear transplants were repeated twice to produce cloned offspring, but it has already been demonstrated that offspring were found to be normal even after 25 repetitions of nuclear transfer ³. Therefore, we believe that there is no long-term adverse effect of this cloning technology on the preservation of genetic resources, which is the primary objective of this paper. However, as you have suggested, this paper may be difficult to appreciate because our experiments combine both FD technology and cloning techniques. Therefore, the following sentence has been added to the text:

Line 267.

“It should, however, be noted that current cloning techniques are not perfect, and it is likely that cloned offspring will have some epigenetic abnormalities due to incomplete reprogramming ⁴. Nevertheless, we have previously demonstrated that the next generation of cloned mice is normal even after nuclear transfer was repeated 25 times ³. In this study, although cloned mice were produced by repeating nuclear transfer twice, those clones were fertile and therefore did not detract from the objective of preserving genetic resources.”

3. To support the conclusion that this cloning technology has no negative influence on development, the authors need to provide more extensive data on the fertility and normal status of various organs in the cloned mice.

In the original manuscript, we mentioned and demonstrated the fertility of the cloned mice (line 170-173: “After maturation, she was mated with a natural male and delivered

the next generation (Fig. 2R). Similar results were observed for male cloned mice derived from FD fibroblasts. These results suggest that the cloned mice derived from FD somatic cells had normal fertility.”)

However, as you have suggested, we have also prepared testicular and ovarian sections from cloned mice and compared them with those of control mice, since, for the purpose of this paper, the state of the reproductive organs is of particular importance. Our results confirmed that there were no obvious differences between these organs in the controls and the cloned mice. We were also able to obtain litters from female cloned mice that were derived from a male donor mouse, proving that the females possess normal fertility. In addition, we now know the lifespan of the first cloned mouse derived from FD somatic cells, and we have added that data as well. The longevity of this cloned mouse was within the normal range. To reflect this additional information, we have revised the manuscript at the lines indicated below:

Line 173-

“Fertility of cloned mice

The fertility of the cloned mice is an important factor in the viability of this technique for preservation of genetic resources. After maturation, we randomly selected nine female and three male cloned mice, which were mated with male and female natural mice, respectively. After 2-3 months, most females delivered the next generation of mice (Fig. 2R, Table S10). These results clearly suggest that the cloned mice derived from FD somatic cells possess normal fertility. In addition, the testes or ovaries of cloned mice were histologically examined to examine the status of the reproductive organs, and compared with control mice, no obvious abnormalities were observed (Fig. S4).

”

Line 192-

“However, as mentioned above, female cloned mice derived from a male mouse were able to successfully deliver pups after mating with a natural male (Fig. S7). These results suggest that even if Y chromosome loss does occur, this technique can still be used to the available genetic resources in extreme circumstances, such as almost extinct species. ”

Figure legend

“Fig. S4. Histological analysis of ovary and testis of cloned mice

Stereomicroscopic picture of ovary from a cloned mouse derived from a male donor. (A) Several growing follicles, including a Graafian follicle, are evident, similar to the control

mouse ovary (B). Similarly, the testis of a cloned mouse (C) demonstrates normal spermatogenesis, as does the control mouse testis (D).”

“Fig. S7. Female cloned mouse derived from a male FD fibroblast delivered offspring after natural mating with a male. Cloned female mice were mated with ICR male mice naturally. Approximately 3 weeks later, the cloned mice delivered offspring, demonstrating that the cloned mice possess normal fertility. “

Reviewer #2 (Remarks to the Author):

Remark for Authors

Review of MS: Healthy cloned offspring derived from freeze-dried somatic cells, by Sayaka Wakayama et al.

The manuscript reports the production, for the first time, of normal and fertile mice following the use of ES cells established from blastocyst cloned using freeze dried somatic cells as a donor. Intermediate steps leading to this achievement, described and discussed in the manuscript, were the characterization of the damage exerted by the freeze dried procedure, both at whole cells and DNA level, including the formation of micronuclei, expression of bold genomic damage, in embryos derived from freeze dried cells. Each step configured itself as a limiting factor, but at the end, through the correction measures adopted by the authors, embryos and embryonic stem cells were derived from dry mouse granulosa and fibroblast cells.

The data are robust, and backed up by thousands of nuclear transfer embryos, and hundreds of ES lines. The demonstration that dry nuclei can develop into normal adults is a remarkable achievement, both for the scientific community and lay audience. Indeed, the presented work is a significant step toward the establishment of alternative, cheaper and safer bio-banking solutions.

We thank the reviewer for their comprehensive understanding of our study and for improving the quality of the study. We have also made appropriate corrections as suggested throughout the manuscript.

Minor Points:

Page 2, line 23:...In this study, we aimed to generate a cloned mice from freeze-dried (FD) somatic cell nuclei, the cells were preserved at -30 °C for up to 9 months. The sentence is not clear: were the cells stored for 9 month at -30 °C prior drying? Please clarify

We apologise for this lack of clarity. We preserved the cells after FD treatment. We have corrected this statement as follows:

Before: “the cells were preserved at -30 °C”

After: “preserved at -30 °C after FD treatment”

Page 2, line 25: and nucleic DNA damage significantly increased when cells were rehydrated. As presented, seem that DNA damage is due to the rehydration, rather that after freeze drying. Please, emend.

We apologise for this mistake. We have corrected this sentence as follows:

Before: “DNA damage significantly increased when cells were rehydrated.”

After: “DNA damage significantly increased.”

Page 2, line 27: Using these cells as nuclear donors for re-cloning, we obtained healthy cloned mice from FD somatic cell nuclei in male or female mice. Better if: Using these cells as nuclear donors for re-cloning, we obtained healthy cloned female and male mice. Please change.

Thank you for this suggestion. We have corrected this sentence as advised.

Page 2, line 32: FD somatic cells will be a superior and safer source. Superior here is inappropriate, fresh cells are “superior” in all respect. Please, change the text accordingly.

We apologise for this misrepresentation and have deleted this sentence.

Page 3, line 60: improving the nuclear transfer procedure. This is not correct, simply, the procedure has been adapted to the kind of nuclei used. Please, change the text accordingly.

Thank you for your careful review. We have revised this phrase as suggested.

Before: “improving the nuclear transfer procedure.”

After: “**adapting** the nuclear transfer procedure.”

Page 7, line 142: ...we could use these ntES cells as infinite donors for serial nuclear transfer. The wording here is not appropriated, please modify the text.

We apologise for the inappropriate wording. We have therefore modified this sentence as shown below:

Before: “ntES cells as infinite donors for”

After: “ntES cells as **better quality and easily prepared donor cells** for”

Page 11, line 240: whereas somatic cell nuclei are not hard,.....better : somatic cells have a more fragile nuclear structure, comparing to spermatozoa.

Thank you for this suggestion. We have corrected this sentence as suggested.

Page 11, line 245: However, this study clearly demonstrated that neither protamine nor half the amount of DNA are essential for successful cloning. The sentence does not add valid elements to the discussion, please remove.

Thank you for your suggestions. We have deleted this sentence, as suggested.

Page 12, line 260: ... will be an important method for the permanent prosperity of humanity. Reads pleonastic. Better to remain within the possible application of the outputs achieved, like the establishment of biobank at low cost, for any purpose, biodiversity, bio medicine, etc.

Thank you for your feedback. We have revised this sentence as follows:

Line 285

“Ultimately, the preservation of somatic cells by FD treatment will be an important method supporting the establishment of alternative, cheaper, and safer bio-banking solutions.”

Reviewer #3 (Remarks to the Author):

Wakayama is an acknowledged expert in the field of mammalian transfer and what he has done here is, I think, valuable. I think that it is essential that the paper is reformatted so that the summary makes very clear the points of the work. That is that the summary should say: what percent of normal adult nuclear transfer was achieved. It should also put the main points of the work in a clear abstract.

We thank the reviewer for their comprehensive understanding of our study and for improving the quality of the study. We have also made appropriate corrections throughout the manuscript. We have now included the success rate of achieving cloned mice in the summary. To clarify our main points, we have replaced the last sentence of the abstract as follows:

Line 24-

“Here, we show that freeze-dried somatic cells can produce healthy, fertile clones,

suggesting that this technique may be important for the establishment of alternative, cheaper, and safer liquid nitrogen-free bio-banking solutions.”

Additionally, I asked a graduate student of mine to review this paper, and below you can find his comments about this paper which I agree with.

My graduate student's opinions: We suggest the editors accept the paper. It is remarkable that the authors reach around 14% efficiency, especially in experiments as delicate as nuclear transfer. It is important that the authors mention the downside of dry-freezing and the significant damages observed in the DNA. We believe the authors should make a specific emphasis on how toxic this procedure is and exclude as successful nuclear transfer the specific cases where complete chromosomes were degraded and/or lost. Independently of whether the nuclear transfer developed into adults, loss of complete chromosome is just another proof of the detrimental effects of dry-freezing. We believe the total number of success cases should not include those where sexual chromosomes were lost.

Importantly, we also observed a substantial amount of self-citations, ignoring important work in the field such as Wilmut et al. 1997 and Gurdon et al. 1958/1962. We recommend that after doing these minor alterations the paper should be published in this journal.

We have taken note of your suggestions and added statements pointing out that chromosome loss in the cloned mice is not considered a successful transfer but could be an acceptable abnormality under certain circumstances.

Line 192-

“However, as mentioned above, female cloned mice derived from a male mouse were able to successfully deliver pups after mating with a natural male (Fig. S7). These results suggest that even if Y chromosome loss does occur, this technique can still be used to the available genetic resources in extreme circumstances, such as almost extinct species.”

Line 240-

“Furthermore, female cloned mice were produced from a male donor mouse, which indicated that this set of techniques may result in chromosome loss in cloned offspring. If the objective is to produce completely identical clones from the donor, these female cloned mice from a male donor would not be considered a success, which would further reduce the ultimate success rate. ”

Line 277-

“In this study, we did not determine the precise point at which the Y chromosome was lost, such as during FD treatment, nuclear transplantation, or the establishment of ntES cells, and this result does represent a failure to completely preserve the genetic resources of the donor mouse. ”

The paper by Wilmut et al. was in fact cited in the original manuscript. However, we did unintentionally overlook the Gurdon et al. paper. We have now cited the following two papers by Gurdon et al. in the revised manuscript.

References-

“15. Gurdon, JB....1958”

“16. Gurdon, JB....1962”

References

- 1 Wakayama, T. & Yanagimachi, R. Development of normal mice from oocytes injected with freeze-dried spermatozoa. *Nat Biotechnol* **16**, 639-641, doi:10.1038/nbt0798-639 (1998).
- 2 Tamashiro, K. L. *et al.* Cloned mice have an obese phenotype not transmitted to their offspring. *Nat Med* **8**, 262-267 (2002).
- 3 Wakayama, S. *et al.* Successful serial recloning in the mouse over multiple generations. *Cell Stem Cell* **12**, 293-297, doi:10.1016/j.stem.2013.01.005 (2013).
- 4 Matoba, S. & Zhang, Y. Somatic Cell Nuclear Transfer Reprogramming: Mechanisms and Applications. *Cell Stem Cell* **23**, 471-485, doi:10.1016/j.stem.2018.06.018 (2018).

REVIEWERS' COMMENTS

Reviewer #1 (Remarks to the Author):

The authors have addressed my concerns in the revised manuscript. I recommend acceptance for publication.

Reviewer #2 (Remarks to the Author):

The Authors have fully addressed my comments, making the manuscript sounder and clearer; nulla osta for its publication in my opinion.